# Estimation of Temperature Recovery Distance and the Influence of Heat Pump Discharge on Fluvial Ecosystems

**Jaewon Jung [1] , Jisu Nam [2], Jungwook Kim [1],\*, Young Hye Bae [3] and Hung Soo Kim [3]**

[1]   Institute of Water Resources System, Inha University, Incheon 22212, Korea; jungjw89@gmail.com
[2]   Gyeonggi-Regional Division, LH, Seongnam 13637, Korea; index77@lh.or.kr
[3]   Department of Civil Engineering, Inha University, Incheon 22212, Korea; yhbaebae@gmail.com (Y.H.B.);
      sookim@inha.ac.kr (H.S.K.)
\*   Correspondence: rlawjddnr1023@gmail.com; Tel.: +82-32-874-0069

**Abstract:** Temperature differences between the atmosphere and river water allow rivers to be used as a hydrothermal energy source. River-water heat pump systems are a relatively non-invasive renewable energy source; however, effluent discharged from the heat pump can cause downstream temperature changes which may impact sensitive fluvial ecosystems. The temperature change associated with heat pump discharge in a river reach was examined using the heat transfer equation in a previous study, but not using models. There were also no studies on the impact of temperature change due to heat pump discharge on river ecosystem elements such as endangered fishes. Therefore, in this study, the water temperature recovery distance of effluent was estimated for a river section in the Han River Basin, Korea, using the heat transfer equation and the Environmental Fluid Dynamic Code (EFDC) model. The water temperature recovery distance was estimated to be 9.7 km using the heat transfer equation and 5 km using the EFDC model in summer. It was also estimated to be 4.5 km using the heat transfer equation and 6.7 km using the EFDC model in winter. Results showed that the water temperature recovery distance results estimated by the heat transfer equation had greater variation than the EFDC model. The water temperature recovery distance could also be used as an objective indicator to decide the reuse of downstream river water. Furthermore, as the river system was found to support an endangered fish species that is sensitive to water environment changes, care should be taken to exclude the habitats of protected species affected by water temperatures within water temperature recovery distance.

**Keywords:** hydrothermal energy; river-water heat pump; water temperature recovery distance; heat transfer equation; Environmental Fluid Dynamic Code (EFDC); Han River Basin

## 1. Introduction

River-water temperature is an important factor with a significant impact on the river environment. This is shown when the community structures of fish and benthic macroinvertebrates are altered by water temperature changes [1]. For example, rising water temperatures narrow the habitation range of cold-water organisms and rapidly increase the growth of specific hot-water fish species, altering the ecosystem balance and distribution of aquatic organisms [2]. Extensive research has focused on the effects of water temperature on river environments [3,4].

River water used in a heat pump system is considered an environmentally friendly source of renewable energy. A river-water heat pump system is an air-conditioning system that uses energy derived from the temperature difference between river water and the air. However, when using river

water as a hydrothermal energy source, the discharge water temperature differs from the intake; thus, the water discharged in the cooling process increases the river water temperature, while the effluent in the heating process lowers it. This temperature change impacts both the discharge site and the downstream section. Therefore, when using river water hydrothermal energy, it is necessary to verify the extent of water temperature changes and their environmental impact on the downstream ecosystem.

River-water temperature is influenced by atmospheric conditions, including air temperature, solar radiation, relative humidity, cloud cover, and wind speed [5], which are related to heat exchange between the air and river [6]. Edinger et al. [7,8] were among the first to conduct comprehensive research on this air–water heat transfer, and defined the concept of equilibrium temperature. They also investigated the discharge temperature half-life from a hydroelectric power plant. Thomann and Mueller [9] and Chapra [10] presented a theoretical method to determine the correlation between thermal energy and water temperature. Furthermore, Sinokrot and Stefan [11,12] modeled river-water temperatures to examine fish habitats under climate change and developed a numerical model to calculate hourly river temperatures using a one-dimensional advection–diffusion heat exchange equation and the finite difference method. Finally, Prats et al. [13] investigated the effects of heat flux on the Ebro River in Spain and defined the recovery distance as the distance at which the temperature difference fell below 0.5 °C following the discharge of hot water. The Korean Ministry of Science and Technology [14] analyzed the distance from the upstream entry point to the equilibrium temperature and determined that rivers with a high flow rate recovered to the equilibrium temperature within 10 km, while low-flow-rate rivers recovered within 1–4 km. The heat transfer equation models the transfer and dissolution relationships of thermal energy between river and air temperature. It is easy to estimate the temperature recovery distance using the equation, but it does not take into account the temperature spread in the transverse and vertical directions.

To simulate the water temperature through hydraulic model, Seo et al. [6] developed a horizontal two-dimensional finite element model to analyze the mixing behavior of heat pollutants, and compared the derived numerical solution with the solution determined using the heat transfer equation. Furthermore, Lee et al. [15] predicted the dispersion range of cool water in a liquefied natural gas (LNG) terminal using the Environmental Fluid Dynamic Code (EFDC) model. Although various studies have investigated the behavior of water temperatures in rivers, no study has applied the model to simulate the range of the effect of river water temperature changes due to effluent from a river-water heat pump.

In addition, changes in water temperature in rivers have been found to have a direct effect on fluvial ecosystems, especially fish habitats [1–4]. Therefore, reviewing the effect of discharged water, which is a by-product of hydrothermal energy using a heat pump, is necessary. To date, there have been no studies examining the effects of heat pump effluent on fluvial ecosystems.

As the demand for river-water heat pump systems is expected to increase steadily, it is necessary to objectively analyze the effects of water temperature changes due to effluent discharged into rivers. Therefore, this study aimed to analyze changes in the water temperature in the river from the effluent discharge point to the point where temperature difference stopped being significant. This distance is called the "water temperature recovery distance". The results of the water temperature recovery distance were analyzed and compared through the simulation of both the heat transfer equation and the EFDC model, which is a three-dimensional hydraulic and water quality model. Lastly, we investigated the habitats of existing endangered fish species within the estimated water temperature recovery distance in the target river. The results of this study are expected to be useful as reference data for evaluating the environmental feasibility of the use of stream hydrothermal energy.

## 2. Methods

### 2.1. Summary of Hydrothermal Energy

Hydrothermal energy can be derived from any water source that has a temperature difference with the air—for example, wastewater, river water, lakes, groundwater, and seawater [16,17]. Generally, hydrothermal resources have a lower-than-air temperature in summer and a higher-than-air temperature in winter, and are mainly used for cooling and heating [18–20].

A river-water heat pump system can concentrate heat by compressing refrigerant gases and then transfer this heat into buildings to provide warmth through heat exchangers. Refrigerant gases cool when the pressure is released and can be exchanged with warmer water from the river. In this way, river water is transported to the heat exchanger, and cooled water is discharged into the river in winter. The river is warmer than the air during winter, so a water-source heat pump is more efficient than a pump using air. The reverse is true during summer, when cool river water is transported to the heat exchanger, and the heated river water is discharged [21].

### 2.2. Area of Study

For a river system to be viable for hydrothermal energy utilization, the stability of the river flow must be considered. Jung et al. [22] identified rivers in Korea with sufficient water flow for hydrothermal applications. To examine possible intake areas, the permissible reference flow rate was calculated and sections of river that exceeded the reference flow rate were examined using geographic information system analysis to determine whether buildings that could benefit from hydrothermal heating and/or cooling were within 1 km of the river. With reference to the results of Jung et al. [22] and based on the ease of data acquisition, the river reach from the Yangpyeong Stage Gauge to the Lower Namhan River Watershed of the Han River mainstream in the central Lower Namhan River Watershed was selected for analysis in this study (Figure 1). This section of the Lower Namhan River Watershed has a channel length of 60.81 km and a watershed area of 2072.72 km$^2$, accounting for 6.02% of the Han River Basin. The average watershed width is 34.1 km, and the watershed shape factor is 1.18. The watershed area of Yangpyeong Water Stage Gauging Station and Lower Namhan River Watershed are 128.9 km$^2$ and 88.5 km$^2$. The main channel length is 25.6 km, and coefficient of roughness is 0.03. Time of concentration is 2.8 hr, curve number is 78.7, and mean elevation of basin is 437.8 EL.m.

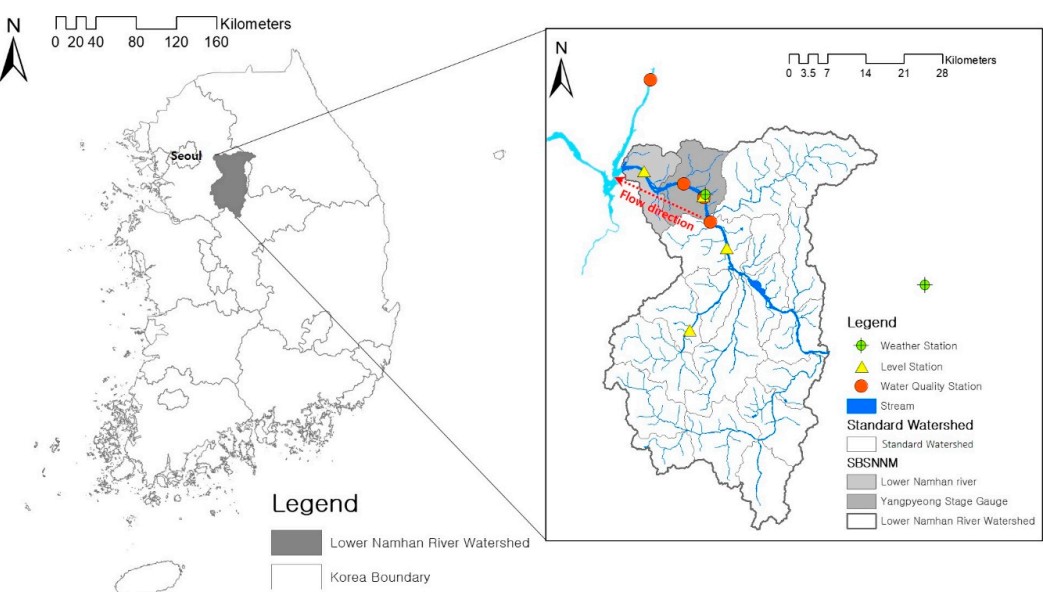

**Figure 1.** The Lower Namhan River watershed, Korea.

*2.3. Estimation of Water Temperature Recovery Distance*

The "water temperature recovery distance" is the distance from the discharge point to the point at which the river water temperature returns to the original temperature of the river. There are two ways to determine the influence of effluent on a river system. This involves simulations using hydraulic and hydrological models or calculation of water temperature distribution using material balance equations [23]. In this study, the water temperature recovery distance was estimated using a heat transfer equation and the EFDC model.

2.3.1. Heat Transfer Equation

A heat transfer equation describing the correlation between river-water temperature and air temperature was used in this study. The water temperature recovery distance was calculated by applying the material balance equation from Edinger et al. [7] and Hwang [24] as follows.

$$T = (T_0 - T_b)\exp\left(-k\tfrac{x}{u}\right) + T_b \ [°C], \ k = \tfrac{K_e}{\rho_w C_w h} \ [s^{-1}],$$
$$x = \tfrac{u}{k}ln|\tfrac{20}{3}(T_0 - T_b)| = \tfrac{\rho_w C_w h u}{K_e}ln|\tfrac{20}{3}(T_0 - T_b)| \ [m], \ |(T_0 - T_b)| \le 0.15 \ °C \tag{1}$$

where $T$ (°C) is the water temperature after the distance $x$, $T_b$ (°C) is the water temperature at the upstream, $T_0$ is the water temperature after the discharge becomes fully mixed with the river water (°C), $x$ is the water temperature recovery distance which is distance until $T_0$ becomes $T_b$ (m), $k$ is the total heat exchange coefficient ($s^{-1}$), $K_e$ is the surface heat exchange coefficient (W/m$^2$ °C), $\rho_w$ is the water density (998.2 kg/m$^3$), $C_w$ is the specific heat of water (4186 J/kg °C), $h$ is the mean river depth(m), and $u$ is the water velocity (m/s).

The boundary condition at the effluent entry point is $T = T_0$, where $T_0$ is the temperature after the river water is mixed with effluent water. Prats et al. [13] estimated the water temperature recovery distance as the distance after which the difference between the mean daily water temperature and mean daily steady state water temperature is <0.5 °C. In this study, the water temperature recovery distance was calculated using Equation (1). This was in accordance with the assumption that the water temperature recovery distance is the distance at which the difference between the river water temperature and the water temperature discharged from heat pump is less than 0.15 °C, as suggested by Prats et al. [13]. Due to lack of research background on the river, the criteria used were based on the objective judgments of the researchers.

In order to estimate the temperature recovery distance, the surface heat exchange coefficient $K_e$ was estimated first. It was estimated using Equation (2), suggested by Edinger et al. [7].

$$K_e = 4.48 + 0.05T_s + \beta f(W) + 0.47f(W) \tag{2}$$

$$\beta = 0.35 + 0.015T_m + 0.0012T_m^2 \ [mmHg/°C], \ T_m = \frac{T_s + T_d}{2} \tag{3}$$

$$f(W) = 9.2 + 0.46W^2 \ \left[W/m^2{\cdot}mmHg\right] \tag{4}$$

where $T_s$ is surface water temperature, $\beta$ is the slope between two points on the saturated vapor pressure curve, $f(W)$ is wind speed function, $T_d$ is dew point temperature, and $W$ is wind speed at 7 m above the water surface.

2.3.2. The EFDC Model

The EFDC, which is an open access model from EPA, is a multifunctional surface-water modeling system that includes hydrodynamic, sediment-contaminant, and eutrophication components [25]. The EFDC model can integrate simulations of density currents (such as water quality), suspended sediments, and saltwater with hydraulic simulations of structures in the water, wet/dry simulations, and material tracking in a three-dimensional hydrodynamic model. The EFDC model

was considered to be suitable for the scenario analyzed in this study [26]. There are five options ("no atmospheric linkage", "full heat balance method", "external equilibrium temperature method", "constant equilibrium temperature coefficient method", and "equilibrium temperature method") for the simulation of water temperature using the EFDC model [27]. These configurations can be set in the card image six (C6) of the EFDC.INP file, which is the main input file for EFDC. In this study, the equilibrium temperature method (CE-QUAL-W2 Method) was used. This method was chosen because it considers advection and diffusion, and easily calculates the water surface heat exchange rate based on the equilibrium temperature and heat exchange coefficient. The CE-QUAL-W2 model is a two-dimensional hydraulic and water quality model developed by the US Army Corps of Engineers and Portland State University. The equilibrium temperature method calculates the water surface heat exchange rate based on the equilibrium temperature and heat exchange coefficient, as shown in Equation (5) [28,29].

$$H_{aw} = -K_{aw}(T_w - T_e) \tag{5}$$

where $H_{aw}$ is the water–air heat exchange rate (W/$m^2$), $K_{aw}$ is the water–air heat exchange coefficient (W/$m^2$·°C), $T_w$ is the water temperature (°C), and $T_e$ is the equilibrium temperature (°C).

For topography data, a grid coordinate file was created using the surface-water modeling system (SMS) program developed by Aquaveo, and the topography information file was used in the EFDC Explorer5 to create the model [30].

### 2.4. Data Collection and Analysis

The weather and level stations used in the analysis are shown in Tables 1 and 2, respectively. The EFDC model requires weather, water level, and flow data for boundary locations. Thus, data were collected from the boundary locations at Ipo Weir, Cheongpyeong Dam, and Paldang Dam.

**Table 1.** Weather station (automated surface observation station) information.

| Station Code | Station Name | Longitude (Degree) | Latitude (Degree) | Elevation above Sea Level (EL.m) | Observation Start (Year) | Observing System |
|---|---|---|---|---|---|---|
| 114 | Wonju | 127.9 | 37.3 | 148.6 | 1971 | ASOS [1] |
| 202 | Yangpyeong | 127.5 | 37.5 | 48.0 | 1972 | ASOS |

[1] ASOS: automated surface observing system.

**Table 2.** Level station information.

| Station Code | Station Name | Longitude (Degree) | Latitude (Degree) | Zero Elevation (EL.m) | Observation Start (Year) | Observing System |
|---|---|---|---|---|---|---|
| 1007697 | Yangpyeong-gun (Sinwon-ri) | 127.4 | 37.5 | 24.3 | 2016 | T/M [1] |
| 1007685 | Yangpyeong-gun (Yangpyeong Bridge) | 127.5 | 37.5 | 19. 6 | 1953 | T/M |
| 1007660 | Yeoju-si (Ipo Bridge) | 127.3 | 37.4 | 26.1 | 2001 | T/M |
| 1015645 | Gapyeong-gun (Daeseong-ri) | 127.4 | 37.7 | 22.6 | 1914 | T/M |

[1] T/M: tele-metering.

To account for the impact of the Ipo Weir, water temperature observation data from 1 January to 31 December 2017, following the weir's construction, were used. The annual average air temperature at the Yangpyeong Weather Station was 12.30 °C and the temperature showed a clear seasonality (Figure 2, grey dotted line). We assumed that river water was used for cooling in summer and heating in winter, and summer and winter were considered to be June–August and December–February, respectively. The summer (winter) average air temperature was 24.88 °C (−1.82 °C).

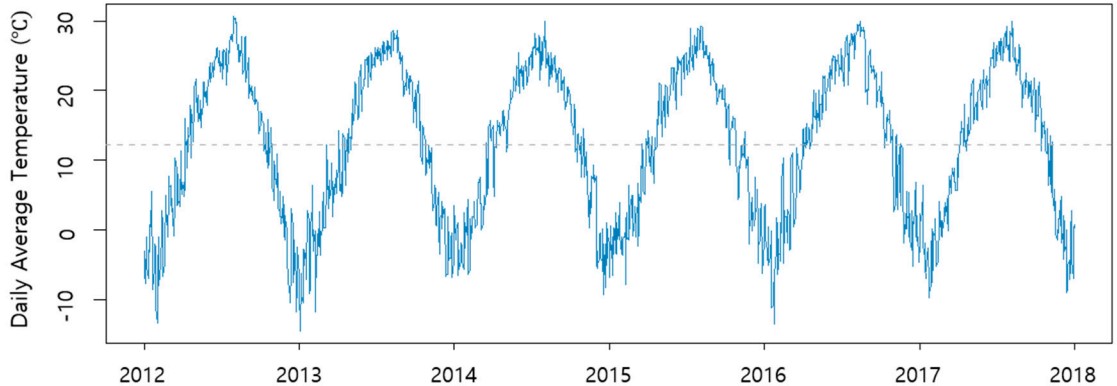

**Figure 2.** Time series of average daily temperatures at the Yangpyeong Weather Station (2012–2017).

When flow rates at the Yangpyeong-gun (Yangpyeong Bridge) Level Station were examined, flow was high during the summer flood season and low during winter. In 2014 and 2015, flow rates were low even in the summer due to drought. The average flow rate of Yangpyeong-gun (Yangpyeong Bridge) was 366.7 m$^3$/s in summer and 126.9 m$^3$/s in winter. Summer flow was approximately three times higher than the winter flow (Figure 3). This was in agreement with the fact that two thirds of the annual rainfall amount falls during the summer season of June to August in Korea.

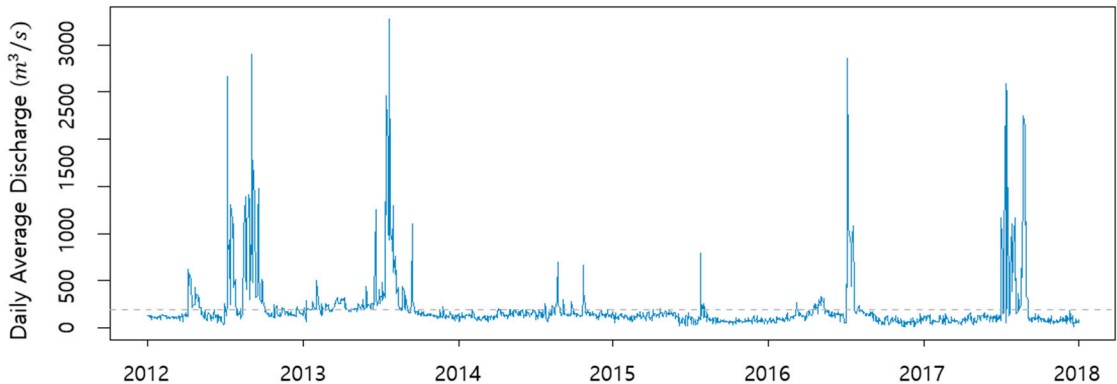

**Figure 3.** Time series of average daily river discharge values at the Yangpyeong-gun (Yangpyeong Bridge) level station (2012–2017).

To account for conditions under which water discharge had the maximum effect, the water temperature recovery distance was estimated using the minimum flow rate during the observation period. Daily and hourly water temperatures were collected from the Yanpyeong, Hongcheon, and Gapyeong Water Quality Stations, which belong to the automated water quality monitoring network of the Water Environment Information System. The Yangpyeong Water Quality Station, which is the closest to the intake and discharge points, was installed in December 2014, and had a data record from April 2015. The annual, summer, and winter average water temperatures were 16.3, 26.3, and 3.7 °C, respectively. Figure 4 compares Yangpyeong Water Quality Station data with the nearby Gangsang water quality monitoring network, which only measures three to four times a month.

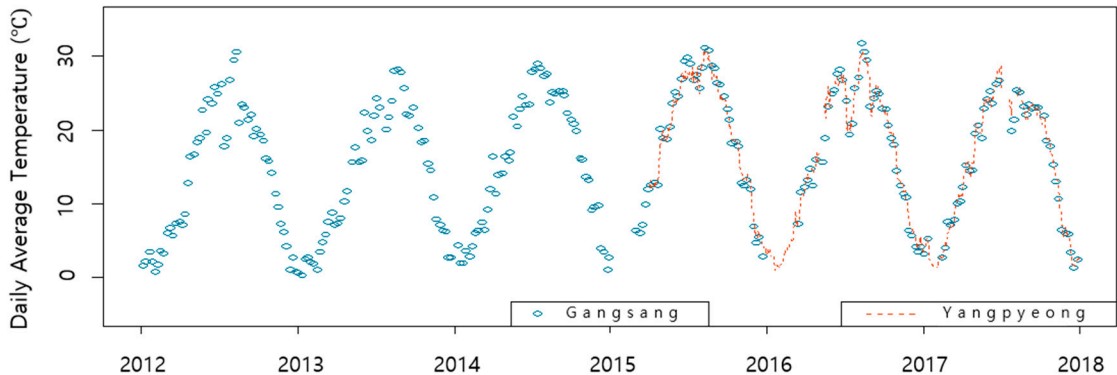

**Figure 4.** Comparison of water temperatures at the Gangsang and Yangpyeong Water Quality Stations.

## 3. Results and Discussion

### 3.1. Heat Transfer Equation Method

Table 3 lists the input data required to estimate the water temperature recovery distance using the heat transfer equation. In order to consider the worst condition for which the maximum temperature recovery distance can be calculated, the following values were used as constants. The average values for each season were used for water temperature, depth, and flow velocity. For the flow rate and surface heat exchange coefficient, the minimum value for each season was used.

**Table 3.** Input data for the heat transfer equation.

| Season | Temperature $T_b$ (°C) | Flow Rate $Q$ (m$^3$/s) | Depth $h$ (m) | Velocity, $u$ (m/s) | $K_e$ (W/m$^2$·°C) |
|---|---|---|---|---|---|
| Annual average | 16.3 | 203.3 | 5.7 | 0.105 | 18.0 |
| Summer average | 26.3 | 366.7 | 5.7 | 0.192 | 23.5 |
| Winter average | 3.7 | 126.9 | 5.7 | 0.006 | 13.1 |
| Summer minimum | 19.9 | 28.1 | 5.3 | 0.001 | 17.8 |
| Winter minimum | 1.0 | 16.7 | 5.2 | 0.001 | 11.8 |

The discharge water temperature was set to 7 °C higher than the river-water temperature during summer, and 5 °C lower during winter. The input and discharge rates were set to 0.64 m$^3$/s . We assumed that discharge water flowed into the existing river and was completely mixed, and that no other weather or environmental changes occurred. Furthermore, we assumed that water depth, flow velocity, and wind speed were constant. That is to say, we assumed that the flow was a fully mixed state because the heat transfer equation cannot analyze three-dimensional case.

The mixed water temperature immediately after entry, $T_0$, was set to the weighted average value of the river flow and discharge. Table 4 shows the seasonal river-water temperature, discharge water temperature, and mixed water temperature immediately after entry. In summer, the mixed water temperature immediately after entry was 0.16 °C higher than the existing river-water temperature, and 0.18 °C lower in winter. Table 5 shows the calculated water temperature recovery distance.

**Table 4.** Seasonal river-water temperature, discharge water temperature, and mixed water temperature immediately after entry.

| Season | River-Water Temperature, $T_b$ (°C) | Discharge Water Temperature (°C) | Flow Rate, Q (m³/s) | Mixed Water Temperature Immediately after Entry, $T_0$ (°C) | Wind Speed (m/s) | Change (°C) |
|---|---|---|---|---|---|---|
| Summer | 26.3 | 33.3 | 28.1 | 26.4 | 1.4 | ▲0.16 |
| Winter | 3.7 | −1.3 | 16.7 | 3.5 | 0.8 | ▼0.18 |

**Table 5.** Water temperature change and water temperature recovery distance results.

| Distance | 0.5 km | 5 km | 10 km | 50 km | 100 km | 300 km | Water Temperature Recovery Distance (km) |
|---|---|---|---|---|---|---|---|
| Summer (°C) | 26.4 | 26.4 | 26.4 | 26.4 | 26.4 | 26.3 | 9.73 |
| Winter (°C) | 3.5 | 3.6 | 3.6 | 3.6 | 3.6 | 3.7 | 4.48 |

### 3.2. EFDC Model

The model grids for simulating the water temperature recovery distance were composed of 89 lateral grids, 234 longitudinal grids, 1484 horizontal segments, and 10 layers in the water column, resulting in a total of 14,840 unit grids. Furthermore, to simulate the target river section in more detail, a model of the target river reach was constructed with 19 grids in the lateral direction (approximately 47.5 m per interval), 310 grids (approximately 75 m per interval) in the longitudinal direction, and 2676 segments in the horizontal direction.

The initial boundary conditions of the water layer were derived from the date of the lowest flow rate in each season during the observation period, including sea-level pressure, air temperature, relative humidity, precipitation, solar radiation, inflow rate, water level, water temperature, wind speed, and wind direction (Tables 6 and 7).

**Table 6.** Weather input data by season.

| Season | Sea-Level Pressure (mb) | Temperature (°C) | Relative Humidity (%) | Precipitation (mm/h) | Solar Radiation (W/m²) |
|---|---|---|---|---|---|
| Summer (6 June 2015) | 1015.0 | 19.6 | 65.5 | 0.00 | 0.85 |
| Winter (20 December 2015) | 1032.2 | −6.2 | 64.9 | 0.00 | 0.50 |

**Table 7.** Hydrological and hydraulic input data by season.

| Season | Inflow Rate (m³/s) | Exit Level (m) | Influent Water Temperature (Hongcheon) (°C) | Wind Speed (m/s) | Wind Direction (10° Intervals from True North) |
|---|---|---|---|---|---|
| Summer | 43.3 | 25.2 | 22.6 | 1.4 | 110 |
| Winter | 16.7 | 25.3 | 2.4 | 0.8 | 200 |

The result of the water temperature was compared to the observed water temperature at Yangpyeong Water Quality Station. As a result, the average simulated water temperature in summer was 24.3 °C, which was about 0.3 °C different from the actual observed water temperature of 24.0 °C. The average simulated water temperature in winter was 1.4 °C, which was about 0.1 °C different from the actual observed water temperature of 1.5 °C. From these results, we concluded that the EFDC model was properly constructed (Table 8).

**Table 8.** The validation results of EFDC.

| Season | Average Simulated Water Temperature(°C) | Observed Water Temperature (°C) | Median Error (°C) |
|--------|------------------------------------------|----------------------------------|-------------------|
| Summer | 24.3 | 24.0 | 0.17 |
| winter | 1.4 | 1.5 | |

The discharge water temperature from the heat pump and its range of influence were calculated for water intake and discharge at 0.64 m³/s. We used the Yangpyeong automated water quality monitoring network site as the discharge location, approximately 250 m downstream from the intake site. Discharge water temperature was 31.0 °C during summer and −3.8 °C during winter, which was 7 °C higher and 5 °C lower, respectively, than the input water.

Immediately after discharge (at i = 156, j = 9; grid no. 2), the water temperature was estimated through vertical grid simulation results for the same lateral number (i = 156) and longitudinal number (j = 9) as those of the discharge site. The simulation period was set to 3 months for both summer and winter, but the front part of the model was removed for the stability of the model. A summer recovery distance simulated on 1 August (123 days) with no intake and discharge was used as the reference value. The longitudinal, lateral, and vertical water temperature recovery distances from the discharge site were then estimated (Figure 5). The summer recovery distance was approximately 4.5 km downstream and 0.5 km upstream (Figure 6). Discharge water impacted the entire river width (300 m) to a depth of approximately 1 m (in 5 m of water).

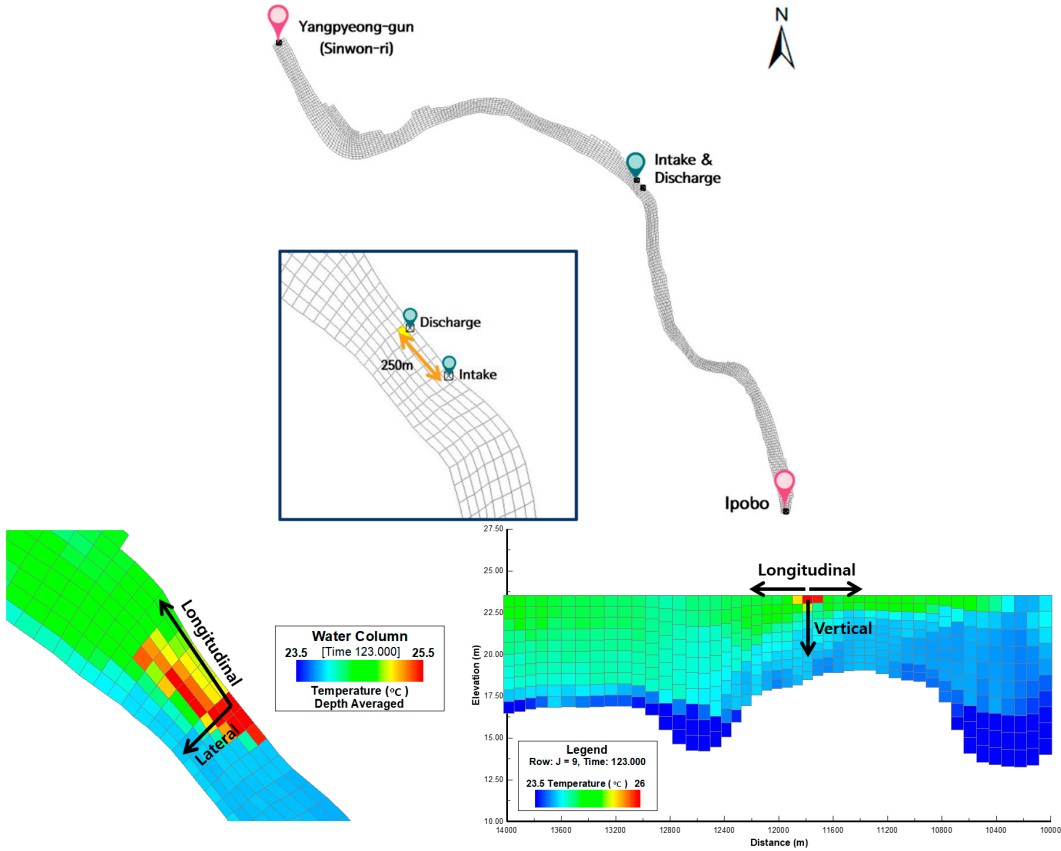

**Figure 5.** Proposed standard for calculating the water temperature recovery distance in longitudinal, lateral, and vertical directions.

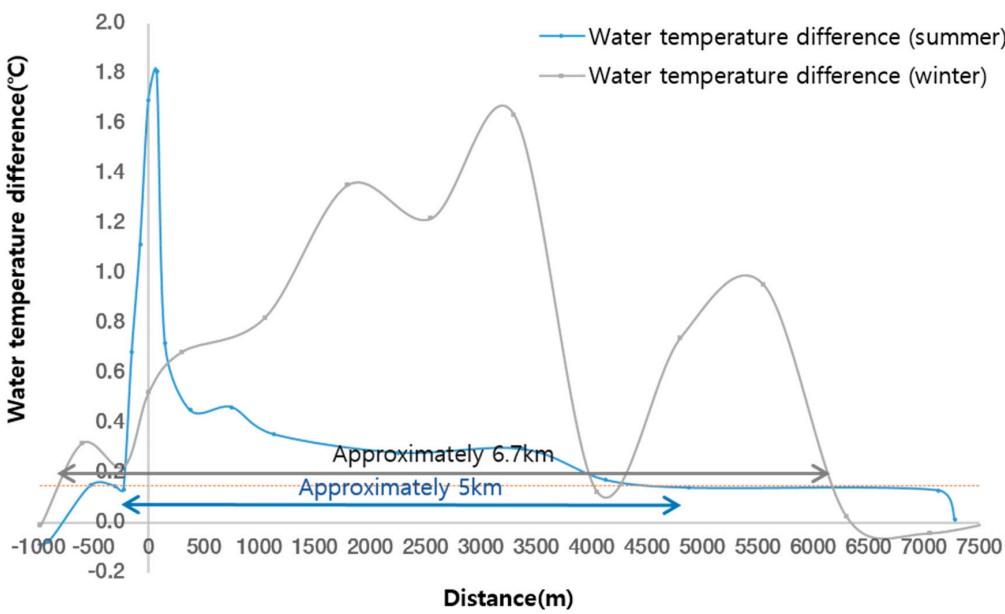

**Figure 6.** Longitudinal water temperature recovery distance estimation graph.

The winter water temperature recovery distance was calculated in the same way as for the summer. The water temperature recovery distance was simulated on 31 January (123 days). The winter recovery distance was approximately 6.7 km (6.1 km downstream and 0.6 km upstream; Figure 6), with a lateral influence of approximately 120 m. The river was impacted to a depth of 3.5 m.

### 3.3. Comparison of Methods

Table 9 summarizes the water temperature recovery distances estimated using the heat transfer equation and the three-dimensional EFDC models. The water temperature recovery distances estimated by the heat transfer equation and the EFDC model were 9.7 and 5 km, respectively, in summer, and 4.5 and 6.7 km, respectively, in winter. The three-dimensional temperature recovery distance of the EFDC model showed that during summer, the longitudinal distance was shorter than the result obtained from the heat transfer equation, but it was wider in the lateral direction during winter. On the other hand, the longitudinal distance during winter was estimated to be longer than the result of the heat transfer equation, while its lateral distance was relatively short. It can be expected that the diffusion of water temperature will take a different form in winter and summer due to the stratification of water and so on. In the case of the heat transfer equation, it was calculated by considering only the longitudinal direction, so it seems that the EFDC simulation result and the difference occurred in simulating the difference between summer and winter characteristics. These differences can be attributed to the fact that the heat transfer equation can be simulated in the flow direction only, and thus it did not account for potential upstream effects. Furthermore, the heat transfer equation model makes a number of assumptions, such as complete and instantaneous mixing, which do not reflect the actual conditions in stream flow. More precise quantitative analysis is possible in the EFDC model, particularly the three-dimensional model and, as such, the recovery distances estimated using this method are more likely to be representative of actual river conditions. However, in this study, there was a limit to estimating the volume of the stream that actually affects it, as temperature recovery distance was estimated using only the longitudinal, lateral, and vertical directions at the effluent discharge point.

**Table 9.** Water temperature recovery distance estimation result by method.

| Season | Heat Transfer Equation (km) | EFDC Model (km) | | |
|---|---|---|---|---|
| | | Longitudinal Direction | Lateral Direction | Vertical Direction |
| Summer | 9.7 | 5.0 | 0.30 | 0.01 |
| Winter | 4.5 | 6.7 | 0.12 | 0.04 |

*3.4. Environmental Impact of River-Water Temperature Changes*

It is important to consider the impact on species vulnerable to water temperature changes within the water temperature recovery distance. In particular, attention should be paid to the existence or absence of habitats for endangered animals within the water temperature recovery distance. Endangered wildlife here refers to species that are determined by the Korean Ministry of Environment as wildlife that may be endangered in the near future if the population is greatly reduced due to natural or artificial threats, or if the current threat is not removed or mitigated [31]. Twenty-five species of fish are currently designated as endangered wildlife in Korea [22]. Figure 7 outlines the status of the endangered animals that inhabit river sections that are viable for hydrothermal applications. It is apparent that *Acheilognathus signifer* inhabits the Yangpyeong Stage Gauge within the target river section analyzed in this study. Acheilognathus signifier is a freshwater fish species indigenous to the Korean Peninsula. The species is highly sensitive to the water environment and populations are declining rapidly because the habitats of their spawning host clams have been disturbed. *Acheilognathus signifer* fertilization and hatching occurs at approximately 22 °C, and the maximum immunity temperature for the species is 28 °C. Therefore, it is possible that heat pump discharge during the summer could impact *Acheilognathus signifer* habitats.

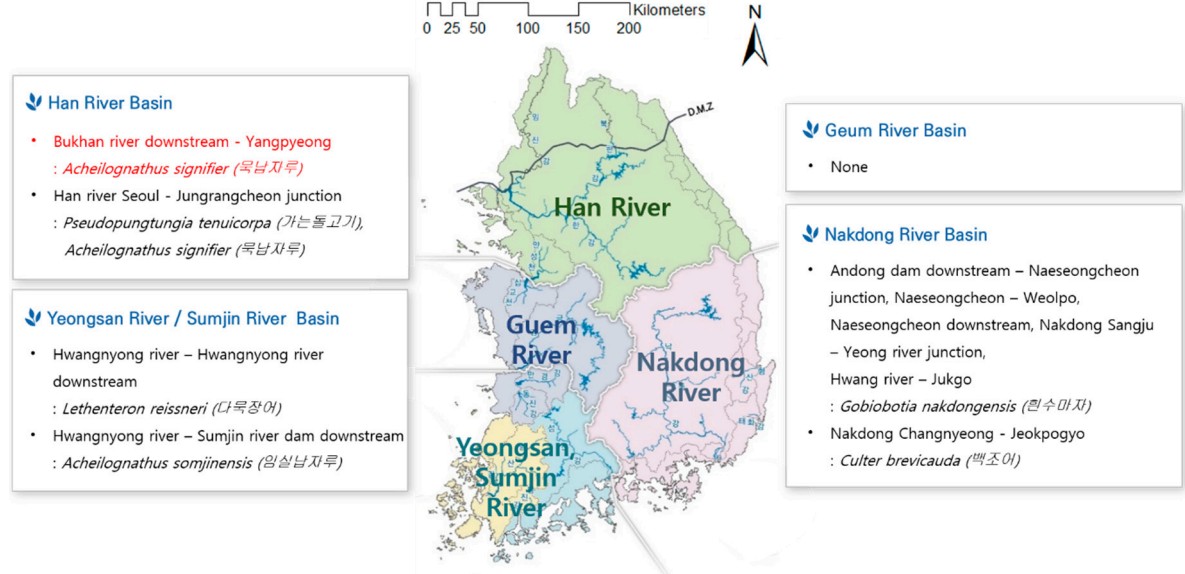

**Figure 7.** Distribution of endangered fish species in Korea (Jung et al. [22]).

Our results indicated that river water hydrothermal installations will impact river-water temperatures at least 4.5 km downstream of effluent discharge and up to a distance of about 9.7 km. Consequently, potential hydrothermal installations should assume a summer water temperature recovery distance of 9.7 km in the target river reach and plan the effluent discharge to avoid impacting water temperatures at the Yangpyeong Stage Gauge.

## 4. Conclusions

River-water temperature changes caused by discharge from hydrothermal heat pumps were investigated to examine the environmental feasibility of river water hydrothermal energy applications in ecologically sensitive areas. A location was selected in the Han River Basin based on a previous study that examined the stability of flow rates for river water heat sources. Changes in water temperature were considered in terms of the water temperature recovery distance—the distance at which the impact of the hydrothermal effluent on river-water temperature is reduced to less than 0.15 °C. Two methods of determining the water temperature recovery distance were used in this study: the heat transfer equation method and the EFDC three-dimensional model.

Discharge water was assumed to be 7 °C higher than the existing river-water temperature in summer and 5 °C lower in winter. The summer water temperature recovery distance was estimated to be 9.7 km using the heat transfer equation method and 5 km using the EFDC model. In winter, the water temperature recovery distance was estimated to be 4.5 km using the heat transfer equation method and 6.7 km using the EFDC model. The heat transfer equation was less reliable, because there were many assumptions made, such as full mixing at same time. The EFDC model could simulated the temperature change between the top and bottom more accurately by dividing the water layer in the depth direction.

The environmental conditions used in this study were chosen to ensure that the simulated discharge water would have the greatest impact on the river environment. Therefore, it is likely that the simulated water temperature recovery distance may have overstated the actual water temperature recovery distance. Notwithstanding, a water temperature recovery distance of 9.7 km suggests that the discharge from a river-water heat pump has no environmental impact beyond 9.7 km from the effluent discharge point. This measure could also be used as an objective indicator for the reuse of downstream river water, and the repeat installation of river-water heat pumps.

When endangered species in the vicinity of the study location were examined, an endangered fish species (*Acheilognathus signifer*) that is sensitive to water environment changes was found to inhabit some sections. Therefore, care should be taken to exclude the habitats of *Acheilognathus signifer* and other protected species affected by water temperatures up to 9.7 km downstream of hydrothermal energy effluent discharge. If the environmental impact of these facilities is inevitable, then substitute habitats should be prepared for the protected species. This study was conducted to examine the environmental feasibility of the utilization of hydrothermal energy, which can be generated by heat pump using a river-water source. Therefore, it is expected that the results of this study could be used as baseline data for environmental feasibility reviews for hydrothermal energy utilization. This could also be beneficial for preliminary consultation of interested parties, including non-government organizations, in countries where river-water hydrothermal energy is lacking, as well as interested residents in the downstream areas. In addition, it is expected that this study could be used to review the reuse of river-water temperature in downstream areas, especially in countries where river-water hydrothermal energy is actively utilized.

**Author Contributions:** Conceptualization, J.J., J.N. and H.S.K.; Data curation, J.N.; Formal analysis, J.J.; Funding acquisition, Y.H.B.; Investigation, J.K.; Methodology, J.J. and J.N.; Project administration, J.J., J.N. and Y.H.B.; Resources, J.N., J.K. and Y.H.B.; Software, J.N.; Supervision, H.S.K.; Visualization, J.K. and Y.H.B.; Writing—original draft, J.J.; Writing—review & editing, J.K. and H.S.K. All authors have read and agreed to the published version of the manuscript.

**Funding:** This research received no external funding or This research and the APC were funded by the National Research Foundation of Korea (NRF) grant funded by the Korea government (MSIT) (No. 2017R1A2B3005695).

**Acknowledgments:** This work was supported by the National Research Foundation of Korea (NRF) grant funded by the Korea government (MSIT) (No. 2017R1A2B3005695).

**Conflicts of Interest:** The authors declare no conflict of interest.

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
