# Peer review of "Estimation of Temperature Recovery Distance and the Influence of Heat Pump Discharge on Fluvial Ecosystems"

_water, doi:10.3390/w12040949_

Round 1

Reviewer 1 Report

The study looks interesting to the readers. Please check my comments below:

  • Abstract: It is always good to include numeric results for readers to initially get obvious insight about the study results.
  • Introduction: Discuss some advantages and disadvantages of this method (Heat Transfer Equation Vs. EFDC model).
  • Introduction: Discuss why you chose to conduct this study using Heat Transfer Equation & EFDC.
  • Line 76: “temperature differential” should be temperature difference.
  • Line 127: Add “a” before non-commercial model. Is it an open access model?
  • Line 140: CE-QUAL-W2 model….. “developed by the US Army Corps of Engineers”. Be advised that Portland State University (The water Quality Research Group) has been the model developer as well. Please check the latest version of the model and add a reference.
  • Please add a reference of the model’s manual.
  • Refer to Fig. 1 in section 2.3.
  • Figure 5 looks very busy, please improve for better quality.
  • Figure 6 could be improved.
  • Lines 178-179: Justification is needed of why summer flow is about three times that in winter.
  • Lines 198-200: Discuss your assumptions in more details.
  • Table 4: Change “Division into Season.
  • If possible, include a figure of the model grid and reference it in section 3.2
  • Lines 247-249: in the observation “The three-dimensional…… during winter” please discuss why.
  • Line 267: “Minister” should be Ministry.

Author Response

Response to Reviewer 1 Comments

Point 1: Abstract: It is always good to include numeric results for readers to initially get obvious insight about the study results.

Answer 1: As what was recommended by the reviewer, the authors included numeric results in the abstract.

[Line 23-27] The water temperature recovery distance was estimated to be 9.7km using the heat transfer equation and 5km using the EFDC model in summer. It was also estimated to be 4.5km using the heat transfer equation and 6.7km using the EFDC model in winter. Results showed that, compared to the EFDC model, the heat transfer equation tended to estimate the bigger variation water temperature recovery distance.

Point 2: Introduction: Discuss some advantages and disadvantages of this method (Heat Transfer Equation Vs. EFDC model).

Answer 2: The heat transfer equation is to compute the transfer of thermal energy that can be occurred by the temperature difference between river water and air. It is easy to estimate the temperature recovery distance through the equation, but it does not take into account the temperature diffusion in the transverse and vertical directions. On the other hand, the EFDC model is a three-dimensional mathematical model that can estimate the temperature recovery distance by longitudinal, lateral, and vertical distances for more accurate quantitative analysis. However, model construction is difficult and a lot of basic input data are required, which increases the probability of error in model results. In addition, it is difficult to verify the 3D simulation results in the water temperature simulation.

Point 3: Introduction: Discuss why you chose to conduct this study using Heat Transfer Equation & EFDC.

Response 3: The authors tried to derive suitable water recovery distance results by comparing the results by the equation with the heat transfer equation and the three-dimensional hydraulic modeling results of the water temperature through the EFDC model.

Point 4: Line 76: “temperature differential” should be temperature difference.

Response 4: As what was recommended by the reviewer, the authors revised it.

Point 5: Line 127: Add “a” before non-commercial model. Is it an open access model?

Response 5: As what was recommended by the reviewer, the authors added “a”. It is open-source software and is a widely used, EPA accepted model. (see Line 163)

Point 6: Line 140: CE-QUAL-W2 model….. “developed by the US Army Corps of Engineers”. Be advised that Portland State University (The water Quality Research Group) has been the model developer as well. Please check the latest version of the model and add a reference.

Please add a reference of the model’s manual.

Response 6: The authors checked and added Portland State University as another model developer. The latest version of the model is ver.4.2 and the manual of it is added as a reference#29.

Point 7: Refer to Fig. 1 in section 2.3.

Response 7: As what was recommended by the reviewer, the authors referred to figure 1 in section 2.2.

Point 8: Figure 5 looks very busy, please improve for better quality.

Response 8: As what was recommended by the reviewer, the authors revised the figure 5.

Point 9: Figure 6 could be improved.

Response 9: As what was recommended by the reviewer, the authors revised the figure 6.

Point 10: Lines 178-179: Justification is needed of why summer flow is about three times that in winter.

Response 10: In general, precipitation tends to be concentrated in summer season in Korea, so river flow also flows more in summer than in winter. (See Lines 199-200)

Point 11: Lines 198-200: Discuss your assumptions in more details.

Response 11: The assumptions presented are assumptions applied to heat transfer equation in general. In the case of the heat transfer equation, since 3-D analysis is impossible, it is calculated using the assumption of a fully mixed state and does not take into account the changes in the external environment.

Point 12: Table 4: Change “Division into Season.

Response 12: As what was recommended by the reviewer, the authors changed it in Table 4, 7 and 8.

Point 13: If possible, include a figure of the model grid and reference it in section 3.2

Response 13: The model grid for simulating the water temperature recovery distance is shown in figure 5 above.

Point 14: Lines 247-249: in the observation “The three-dimensional…… during winter” please discuss why.

Response 14: It can be expected that the diffusion of water temperature will take a different form during winter and summer due to water stratification. The heat transfer equation is calculated by considering only the longitudinal direction, since EFDC simulation results show a significant difference between summer and winter characteristics.

Point 15: Line 267: “Minister” should be Ministry.

Response 15: As what was recommended by the reviewer, the authors changed it.

Reviewer 2 Report

The Study area section should go first before Method section

What is the innovative points of this study?

What do you think about the using combination of observation data and satellite data to monitoring the water temperature at larger scale instead of limited to use observation station?

It seems only 2 weather stations are used for the model and validation? it is good enough?

Is there influence of rainfall on water temperature? and it is considered in your model?

In the Introduction Section the authors wrote: "there is few which have examined the environmental impact of river-water heat pump discharge" Line 65-66, so which environmental factors the authors consider in the analysis and how it impacts? such as Chlorophyll I only found the climate factors like wind, seasonal change...

The manuscript needs to be enhanced, especially the introduction to enhance the objectives and innovative points and conclusion, an how the result was validated?

Author Response

Response to Reviewer 2 Comments

Point 1: The Study area section should go first before Method section

Response 1: As what was recommended by the reviewer, the authors changed the order of section 2.2 and 2.3.

Point 2: What is the innovative points of this study?

Response 2: The concept of water temperature recovery distance was applied to examine the utility of river water hydrothermal energy. In addition, the results of the water temperature recovery distance were derived and compared through the simulation of both the heat transfer equation and the EFDC model. Lastly, we investigated the habitats of existing endangered fish species in the target river and reviewed their impact on the ecosystem.

In the field of hydrothermal energy, there have been no studies using a model such as EFDC to find the temperature recovery distance. In particular, there have been no research cases examining the effect of heat pump discharge on the ecosystem of a river.

Point 3: What do you think about the using combination of observation data and satellite data to monitoring the water temperature at larger scale instead of limited to use observation station?

Response 3: If there is available satellite data for monitoring the water temperature, the authors believe that we can compare satellite data with observations for more reasonable result. However, we may consider the reviewer’s comments in further study.

Point 4: It seems only 2 weather stations are used for the model and validation? it is good enough?

Response 4: In this study, since the temperature changes at two points upstream and downstream are viewed, analysis is possible with only two points.

Point 5: Is there influence of rainfall on water temperature? and it is considered in your model?

Response 5: We assume that the heat pump water is discharged to the river during the period of no rain.

Point 6: In the Introduction Section the authors wrote: "there is few which have examined the environmental impact of river-water heat pump discharge" Line 65-66, so which environmental factors the authors consider in the analysis and how it impacts? such as Chlorophyll I only found the climate factors like wind, seasonal change...

Response 6: When the discharged water is returned to the river after using hydrothermal energy, analysis in the possibility of secondary environmental load is necessary. However, the said secondary load tends to be insufficient. When the effluent which has a temperature different from the existing water is discharged, water temperature recovery distance from the inflow point is measured.

Also, changes in water temperature on the river is calculated. Since the water quality together with flow rate and depth are major concerns for fish habitats, and since water temperature is an important indicator of water quality, analysis done in this study is beneficial in the environmental field.

Point 7: The manuscript needs to be enhanced, especially the introduction to enhance the objectives and innovative points and conclusion, an how the result was validated?

Response 7: As what was recommended by the reviewer, the authors enhanced the introduction.

Round 2

Reviewer 2 Report

It is fine by me at current version. However, need to check once again over the manuscript and correct the mistake such as  9.7km-->  9.7 km similar for other

habitats[1-4]. --> need a space before [

Author Response

Thank you very much for comments.

As what was recommended by the reviewer, the authors correct the mistakes over the manuscript.
